# Wireless, Material-Integrated Sensors for Strain and Temperature Measurement in Glass Fibre Reinforced Composites

**DOI:** 10.3390/s23146375

**Published:** 2023-07-13

**Authors:** Lukas Bertram, Michael Brink, Walter Lang

**Affiliations:** 1Institute for Microsensors, Actuators and Systems (IMSAS), University of Bremen, 28359 Bremen, Germany; 2BIBA—Bremer Institut für Produktion und Logistik GmbH, 28359 Bremen, Germany; bri@biba.uni-bremen.de

**Keywords:** wireless, temperature, strain, sensor, composites, FRP, RFID, structural health monitoring

## Abstract

Fiber reinforced plastics (FRP) offer huge potentials for energy efficient applications. Special care must be taken during both FRP fabrication and usage to ensure intended material properties and behavior. This paper presents a novel approach for the monitoring of the strain and temperature of glass fibre reinforced polymer (GFRP) materials in the context of both production process monitoring and structural health monitoring (SHM) applications. The sensor is designed to be integrated into GFRPs during the production process, and the sensor concept includes possibilities of automated placement during textile layup. To minimize sensor impact on GFRP integrity and to simplify vacuum setup and part handling, the sensor operates without the need for either wires or a battery. In the first sections of this work, sensor concept, design and prototype fabrication are presented. Subsequently, it is shown how the sensors can be used for flow front monitoring and cure estimation during GFRP production by measuring local resin temperature. The resulting specimens are then characterized regarding strain measurement capabilities, mechanical influence on the host component and overall system limitations. Average strain sensor accuracy is found to be ≤0.06 mm/m, while a maximum operation temperature of 126.9 °C and a maximum reading distance of 38 mm are measured. Based on a limited number of bending tests, no negative influence of sensor presence on breaking strength could be found. Possible applications include structural components, e.g., wind turbine blades or boat hulls.

## 1. Introduction

Components and structural parts made from Fiber Reinforced Polymers (FRPs) have been increasingly gaining in relevance in recent years, as they can combine unique ratios of weight, strength and stiffness in a single material [1]. As opposed to metals, FRPs can achieve similar values for stiffness and strength with only a fraction of the weight. Regarding durability and fatigue resistance, FRP materials have been found to outperform similar components made of metal, e.g., as presented in [2,3,4]. Moreover, as FRP materials are or can be anisotropic by design, it is possible to construct mechanical parts according to application-specific loading scenarios, allowing for an overall reduced input of material, once again saving energy and resources. In the context of rising prices for energy and resources, FRP materials are therefore especially interesting for lightweight construction applications, aiming to reduce energy consumption during usage and for the reduction in overall carbon footprint.

This is possible as FRPs are a combination of two inherently different materials: one material is the fibers that are light, usually strong and able to bear high loads, but due to their nature are not directly usable for construction. The other material is the polymer matrix that encloses the fibers, thereby fixing them in position and protecting them against environmental influences [5]. This way, the forces generated by the application are optimally directed into the fibers, and the resulting part is able to withstand much higher loads than either the fibers or matrix on their own.

Despite all their unique advantages, FRP materials also have their respective disadvantages and require a complex and carefully controlled fabrication process to achieve the outstanding properties just described.

### 1.1. Conducted Work

This paper presents a novel sensor concept aiming to improve two major aspects for FRP materials: fabrication monitoring and load monitoring. It utilizes Radio Frequency Identification (RFID) technology to make a battery-less, wireless sensor system for easy integration into Glass Fiber Reinforced Polymer (GFRP), which can be utilized to locally monitor resin infusion and resin cure during fabrication, thereby improving the process and quality control while offering additional advantages for component handling and maintenance. Additionally, during component application, it can be used in a Structural Health Monitoring (SHM) context to wirelessly monitor mechanical loading and local temperature inside the laminate.

It incorporates a temperature-compensated, full bridge strain gauge for bidirectional strain measurement, a precise digital temperature sensor, and a combined micro-controller and RFID-Transponder for communication, power supply, data acquisition and data storage. For usability in increasingly automated production environments, sensor design specifically targets the easy applicability for automated sensor positioning, e.g., via robot. Standardized RFID communication standards (ISO 14443, 13.56 MHz, [6]) are used in order to enable wide compatibility. In order to allow for easy and economical producibility, only off-the-shelf components were used and the sensor substrate (i.e., Printed Circuit Board (PCB)) was fabricated using a standard industrial process.

Experiments show that the system presented can be used for monitoring of the resin flow front in a Vacuum Assisted Resin Infusion (VARI) process while providing information about the progression of the resin cure. The sensor is able to withstand conditions present during the process used and bending experiments show that the bi-axial measurement of the mechanical strain is possible with an average error of 0.059 mm/m. In destructive tests with sensor-integrated FRP specimens, a negative influence of the sensor presence on the bending strength of the host material could not be found, though further research is necessary to investigate the effect of the inlay on the material in greater detail. For the experiments conducted, it can be concluded that the system is able to fulfill its intended purpose, while optimization of sensor geometry, reading distance and strain sensor accuracy is possible.

### 1.2. Structure of This Paper

Prior to a description of the presented sensor itself, motivation for its design and corresponding problems in FRP fabrication and application are described in Section 1.4, Section 1.5 and Section 1.6. After giving the motivation and state of the art in Section 1, sensor concept and fabrication are presented in Section 2. Experiments conducted for sensor characterization are described subsequently in Section 3, while Section 4 presents in detail the corresponding results. Section 5 discusses applicability and open questions for industrial usage of the system presented. Finally, concluding remarks about this work are given in the last section.

### 1.3. Fabrication Monitoring

Fabrication techniques for FRP are numerous and the resulting product properties vary depending on the process used [5]. The overall principle is usually similar—in order to combine the advantageous properties of the two ingredients, the fibers need to be surrounded by the matrix material and arranged into the desired shape. The latter usually implies that the matrix needs to be fluid at some point, in order to enclose the fibers and adjust to the shape. Some techniques use preimpregnated textiles (“prepregs”), i.e., textiles covered in a matrix material before arrangement in the mold, but most techniques relevant for structural parts combine fibers and a matrix after placement in the mold [5].

#### 1.3.1. Infusion

Before the matrix is introduced in fluid form, dry fiber textiles are arranged in the desired shape and thickness and fixed into position. After insertion of the textiles, resin infusion is usually supported by the application of a vacuum inside the closed mold, so the mold needs to be sealed and evacuated before the infusion starts. Evacuation also prevents the formation of holes and voids caused by local pockets of residual air. Additionally, the fluid resin itself is degassed in advance, also using a vacuum, to let any dissolved gas and bubbles rise to the surface and leave the resin. For single-sided molds, as used in VARI processes, a flexible vacuum foil forms the secondary part of the mold. The resulting pressure difference between the inside and atmospheric pressure forces the resin into the mold, while also compressing the fibers to achieve a high fiber density and consequently a higher part strength. As the material properties of the fabricated part are strongly influenced by various process parameters, the relevant parameters require detailed control. The most important parameter for FRP fabrication is temperature [7,8], as it decides both resin behavior (e.g., viscosity) and the behavior of the hardening reaction. Before resin infusion starts, mold and resin temperature are therefore adjusted to achieve the desired resin viscosity and flow rate, and thereby obtain complete impregnation of the fiber material, while the temperature still needs to be low enough to not start resin hardening [8]. Temperature consequently needs to be controlled carefully during infusion, as it affects the progression of the resin flow front through the fiber material. Incomplete impregnation can lead to the formation of holes or voids, which degrade the loading capabilities of the produced part. In some cases, impregnation can to some degree be monitored visually. e.g., if transparent vacuum foil and resin are used and the part/mold geometry is sufficiently simple. However, especially for complex shapes and thick layups, visual flow front monitoring is either impossible or imprecise. In this context, the inhomogenous distribution of temperature can therefore adversely influence mechanical part properties. As temperature distribution can vary throughout the mold and layup thickness due to several reasons—e.g., non-uniform heating or unintended heat dissipation—monitoring and control of processing temperature is a challenging task and holds huge potential for process optimization and quality control. Direct monitoring of the flow front and impregnation could therefore also aid in achieving higher product quality, while simultaneously reducing wastage and energy consumption [9].

#### 1.3.2. Curing

After infusion, in order to make the final part or structure, the fluid matrix needs to become solid again. This hardening process is generally known as curing, and also needs to be controlled carefully to achieve the desired mechanical behavior of the resulting structure. Most curing reactions are exothermic, i.e., produce heat, in turn creating another influence on temperature distribution throughout the layers [8]. As the curing reaction rate is also dependent on temperature, this can create a feedback loop, potentially creating local temperature maxima, which can degrade the quality of the finished part [10]. To prevent these temperature peaks, the curing temperature is usually chosen to be comparably low, resulting in long curing times. As the cure state itself usually is not monitored directly, and especially not inside the laminate, curing progress is often subject to estimation. To ensure sufficient resin cure throughout the whole laminate, curing times are conventionally chosen to be longer for safety reasons, resulting in high cost both in terms of production and energy [7]. The time needed for curing is therefore a major part of production. According to [11] curing takes up to 88% of the overall processing time, presenting a huge potential for time and energy savings. Material-integrated temperature monitoring can consequently help shorten curing times as, on the one hand, processing temperature can be controlled more efficiently. On the other hand, if curing kinetics (i.e., correlation of reaction progress and time) of the specific resin material are known, monitoring of the local temperature inside the laminate can allow for the estimation of the curing degree [7], in turn reducing processing time and cost by enabling the cure-state-based termination of the heating.

#### 1.3.3. State of the Art—Fabrication Monitoring

As described in the prior paragraphs, monitoring of FRP fabrication holds huge potential for process optimization and quality control, which is why it has been the subject of manifold research. Several general principles have emerged, the most prominent of which will briefly be summarized in the following paragraph.

A very precise technique uses measurement of the heat generated by the curing reaction for the calculation of the degree of the cure. It is called Differential Scanning Calorimetry (DSC), and is the state of the art for detailed cure characterization. Though very precise, it is not applicable for in-line process monitoring due to its general principle, as it requires a special reaction chamber and usually only uses small amounts of resin [7,12].

The most prominent technique is called Dielectric Analysis (DEA). It is able to yield good insights into the curing status by measuring changes in the electrical permittivity of the polymer that are caused by the reduced mobility of the charge carriers inside the matrix material [7,12,13]. As of now, DEA techniques require complex and cost-intensive measurement equipment. For actual in-line usage in FRP fabrication, cure measurement using DEA is restricted to surface-based sensors, as the dielectric sensors need direct contact to resin, i.e., the component surface [7,14].

As pressure plays an important role for infusion and resin flow through the fibers, pressure sensors can also be used for the monitoring of flow front and impregnation, as the local pressure changes with the resin arrival, especially in a vacuum [8]. Wired piezoresistive pressure sensors were used by [15] to measure the pressure gradient during a VARI production process of wind turbine blades. Similarly, ref. [9] used wired pressure sensors during FRP fabrication, measuring the local matrix pressure during infusion, to allow for curing initiation only after full impregnation.

For implementation in fiber composites, usage of optical fibers for fabrication monitoring is also a rather obvious possibility, as the intensive research and the numerous publications about FRP fabrication monitoring show. Primarily, these utilize sensors based on Fiber Bragg Gratings (FBGs) (e.g., [16,17,18]). With these systems, monitoring of both flow front and cure status is possible, and, depending on the system, later usage of the FBGs for SHM of the host component is possible (see also Section 1.4). A major downside of systems using FBGs is the sensitive nature of the fibers themselves, making them fragile, complicating handling and resulting in a laborious and therefore cost-intensive installation process [8]. Moreover, the measurement equipment necessary for the interpretation of light signals is complex and comparably expensive.

The simplest and most straightforward approach for FRP fabrication monitoring is the continuous measurement of temperature [7,8]. Therefore, it is usually employed in at least a very basic form (e.g., placement of wired temperature sensors on the mold surface or in the resin reservoir). However, as mentioned above, surface-based temperature measurements can often not yield sufficiently precise information for accurate process monitoring, due to inhomogeneous temperature distribution throughout the laminate. Especially as temperature can vary over laminate thickness for parts with many textile layers, local temperature inside the laminate can very much differ from the surface temperature. This was proven by [19]. They integrated wired temperature sensors both into a FRP layup and onto the mold surface for a VARI process, and showed that surface-based measurement was not able to provide the same amount of information as the integrated sensors. Comparatively, experiments in [20] show that flow front monitoring is possible using wired, laminate-integrated thermocouples for non-metallic molds, while [21] showed that resin flow monitoring is possible with wired thermocouples distributed over carbon fiber preforms for usage in a VARI process. This shows that material-integrated temperature measurement provides superior information as compared to measurement on the surface.

### 1.4. Load Monitoring

The second major motivation for this work is the monitoring of mechanical loads that the FRP material is subjected to during application. Over a component’s life cycle, these can have an influence on its structural health and integrity, for example, due to wear symptoms or material fatigue. Additionally, a component may witness extraordinary loads caused by unforeseen scenarios like extreme weather or human failure. Though this is the case for components of every structural material, and therefore no new problem for design engineers, it is of a higher relevance for structural parts made from FRP. This is caused by a major disparity in the different materials’ reactions when subjected to mechanical loads—while a component made of metal will very likely show a dent or other visual mark following, e.g., a high impact, FRP materials which usually do not show any visual sign of wear, even though the experienced load might have been critical, damaging the part [22]. Additionally, failure mechanisms of FRP structures are difficult to predict, as due to their anisotropic nature, fatigue behavior is dependent on several factors (e.g., layup sequence, fiber directions, materials etc.). Local delamination or broken fibers, which are a common result of overloads in FRP, can frequently be invisible from the outside [22,23,24]. Other than common construction materials such as metals, a component may therefore look unharmed during visual inspection, but may already be damaged internally. Even though these damages might not directly lead to part failure, they can be a cause for unexpected failure during later usage of the structure, with potentially dangerous or fatal outcomes [22].

To reduce the associated risk, techniques helping with the assessment of both the amount of experienced loads and the actual structural health of a component are of high interest for FRP construction and application. These are referred to as SHM technologies. In a more general sense, SHM describes principles for sensing and recording data about the health and or condition of a structure in an automated way, so that it can be used with devices to detect critical scenarios and consequently the damages and risks to the structure [25]. Apart from improvement of FRP reliability and safety, SHM technologies offer further potential advantages, e.g., maintenance on-demand, or usage of SHM sensor data for optimization of the structure or machine function itself.

One of the basic ways to monitor a structure’s load is the quantification of mechanical strain [26]. It is defined by [26] as the ratio of the change in geometrical length Δl relative to the original length $l_{0}$: (1)$$\epsilon =\frac{\Delta l}{l_{0}}$$

In order to quantify strain in the context of SHM, a multitude of principles have been investigated, as comprehensively reviewed in [22,27]. The two most prominent techniques shall briefly be described in the following paragraphs.

#### 1.4.1. Strain Gauges

A very basic principle for strain quantification is the usage of strain gauges, which change resistance analogous to material strain by incorporating a thin and preferably long structure of a piezoresistive material (e.g., a metal) onto a flexible surface (see Figure 1).

One of the advantages of strain gauges is their simple and well-known measurement principle, resulting in a very straightforward circuitry for readout with low power consumption, depending on gauge resistance. This makes strain gauge application very easy. Measurement errors caused by differences in the thermal expansion of the strain gauge and host material can be eliminated by fitting the thermal expansion coefficient to the intended material. Combined with different possible circuit configurations, very high measurement precision and also insensitivity to environmental changes can be achieved with strain gauges [28]. Usually, strain gauges are applied to the component surface with specialized glues to achieve good coupling and therefore transfer of strain into the gauge.

This surface application exposes the gauge to environmental influences, such as water or mechanical impacts, making necessary a protective encapsulation around the gauge and the connecting wires.

For application with FRP, these downsides could be eliminated by [29,30]. They embedded strain gauges into the FRP with good results, thereby showing that material integration of strain gauges into FRP is possible, obviating the need for a glued connection and encapsulation, while also protecting the gauge from environmental influences. Additionally, it could be shown that strain gauge integration can yield a higher sensitivity compared to glued strain gauges due to better mechanical coupling, while also giving insights into strain distribution throughout the component [29].

#### 1.4.2. Fiber-Bragg-Gratings

As with fabrication monitoring, techniques employing FBGs for usage in FRPs have been the subject of intensive research [22,31]. On the one hand, these systems combine several advantages: as optical fibers can be embedded in the textile, near non-intrusive integration into FRP materials is possible, reducing the impact of the sensor presence on the host components’ structural integrity. As integration is done prior or during FRP fabrication, FBG systems can enable the in situ monitoring of relevant processing parameters (see Section 1.3.3), thereby improving process and quality control. During component usage, they allow the in-depth measurement of multiple parameters, e.g., both strain and temperature, as opposed to conventional piezoresistive strain sensors. On the other hand, FBG-based systems also have several severe disadvantages. As already stated in Section 1.3.3, specialized analysis equipment is required in order to make use of the optical fibers. This makes the overall system comparably expensive [22]. Moreover, optical fibers are very sensitive, requiring very careful handling during FRP production and usage [24,31,32]. This in turn increases the cost and makes SHM systems based on FBG feasible only for some scenarios, e.g., involving high-risk structures. In the context of the possibilities for automated system application during FRP fabrication—by, e.g., a robot—this fragility of the optical fibers is a disadvantage.

Further references and details on sensors for SHM techniques can be found in the corresponding literature (e.g., [22,27]).

### 1.5. Problems with Wired Sensors for Component and Fabrication Monitoring

As shown in the preceding paragraphs, numerous techniques and principles have been developed and are employed for monitoring both FRP fabrication and the structural health of FRP components. A mutual property of nearly all the systems mentioned is the usage of wires for the interconnection of the sensors and the measurement devices for transmission of energy and measurement data. Compared to surface-based measurements, material-integrated sensors are gaining in relevance due to higher overall sensitivity and superior protection of the sensor system against exterior influences [27]. For material-integrated sensors, several major problems arise when using wires:During FRP production, the integration of sensors and wires is a complicated and therefore time-consuming process, especially for sensitive sensor elements, and usually has to be done manually, resulting in high costs [8]. Apart from the sensors themselves, wire presence thoroughly complicates the setup of the vacuum seals, as the wires need to be lead though the vacuum seal while still ensuring air tightness. Though this is possible, it is expensive and also prone to errors, potentially compromising vacuum buildup and thereby quality of the infusion process. Additionally, wires can negatively affect the flow front during infusion by creating flow channels along their length.Apart from fabrication-related aspects, wires of integrated sensors also need to be taken special care of during the FRP’s lifetime, as they are very sensitive, especially in comparison with the FRP material itself. This is mainly due to the general arrangement of protruding wires and the surrounding matrix where the wires exit the FRP, that is very likely to create shear stress to the wires, possibly leading to breakage. This is compounded by the difference in the two materials’ Young’s modulus and different thermal expansion coefficients. Once a wire is broken, the sensor system is often rendered useless, as exchange of the integrated sensor is mostly either impossible or economically unreasonable.Regarding the fatiguing behavior of wired, sensor-integrated FRP samples, ref. [33] observed that failure often occurred at the positions where the wires exited the matrix, supporting the conclusion that wire presence can degrade the fatigue behavior of FRP.

### 1.6. Solution: Wireless Sensors for FRP Integration

To circumvent the problems associated with wired sensors, recent research has investigated the development of wireless sensors for integration into FRP.

In order to supply a wireless sensor system with energy, the most straightforward idea is usage of a battery. For material integration, this approach is not feasible, as batteries are comparatively large, thereby negatively influencing the host material, and also have a finite lifetime.

These two constraints make wirelessly supplying the sensor system necessary. This idea is the basis for a prominent technology for wireless transfer of both energy and data, which is called RFID. Though the term RFID encompasses numerous distinct subtechnologies and corresponding standards, the general principle of electromagnetically coupled reader and a to-be-read tag is universal and can also be used for sensoric tags, which are sometimes consequently called sensor tags. Sensor systems comprising their own power source are usually called active, while systems powered remotely are usually called passive, as they rely on wireless power transmission to function [34]. Passive systems are therefore employing a principle commonly known as energy harvesting, i.e., collection of energy from their immediate environment. Though several principles for energy harvesting exist (e.g., heat, vibration, light etc.), most sensoric RFID systems use electromagnetic fields for energy supply, which are supplied by the reading device [34]. The electromagnetic field induces a voltage in an antenna structure connected to the sensoric system, which in turn supplies the sensor.

Two basic principles for the implementation of passive sensor tags have emerged:**Sensoric antennas**: the first and most widely used concept makes direct use of the antenna itself, i.e., it employs the antenna as a sensing element [34,35]. This is possible, as the antenna is subjected to the immediate environment of the sensor and is therefore influenced by it (e.g., changes in resonance frequency due to temperature). This allows for very simple sensor design and therefore low production cost, which is the greatest advantage of sensoric antenna systems. The greatest drawback of sensoric antenna systems is the high susceptibility of the measured entity to exterior influences. As sensitivity of the antenna to external influences is actually part of the system concept, cross-sensitivities of the antenna can degrade system performance [35]. For simple measurement task though, sensoric antennas can be applicable. Regarding usage of sensoric antennas for SHM, ref. [35] comprehensively reviews different concepts and applications.**Dedicated sensing element**: the second principle also uses an antenna, but only for energy supply and communication, while sensoric data are gathered via a dedicated sensing element. In order for this to work, further circuitry is necessary in the sensor, which does the actual measurement acquisition, power control and communication. If designed with sufficiently low power consumption, basically all conventional measurement electronics could theoretically be used on a sensor tag, allowing for much more complex sensing functions.

#### Wireless Sensors—State of the Art

For integration of passive wireless sensors into FRP, a limited amount of prior work has been published to date. Implementing the sensoric antenna approach with a custom reading unit, ref. [36] developed a rigid PCB for wireless, substrate-integrated measurement of epoxy resin cure and local temperature. Similarly, ref. [37] built an LC-resonator circuit using a sensoric antenna and a planar capacitance for integration into FRP to measure curing progress via changes in the resonant frequency.

For using dedicated electronics, as opposed to sensoric antennas, ref. [38] proved that wireless power transfer into FRP is possible via the inductive coupling of a reader coil and an antenna coil, and that the amount of transferable power can be sufficient for basic low power electronics. Ref. [39] proved that embedding passive wireless sensors into FRP is generally possible and usable for process monitoring of FRP production. In their experiments, they were able to measure local pressure during resin infusion in a VARI process, allowing for monitoring of the flow front progress. In another experiment [40], they measured both temperature and resin pressure during FRP infusion and curing, again monitoring the flow front, but the sensors dropped out as temperature increased during curing. The sensors were therefore only usable during infusion and not for cure monitoring or measurements in a SHM context.

A different approach was taken by [41,42], who, in two experiments, integrated several commercially available RFID transponders into GFRP and measured the Received Signal Strength (RSSI) of the transponders to correlate with that of the infusion and curing progress. In the employed VARI process, they were able to identify the arrival of the flow front, but signal changes during curing were too small to allow for correlation with the curing status.

## 2. Sensor Design and Fabrication

After the motivations for this paper have now been presented in the prior section, this section describes the concept and fabrication of the sensor designed to solve some of the above problems, giving details on the circuitry, the electronics used, micro controller programming and the steps needed for the physical buildup of the final prototype. The sensor electronics are based on results presented in [43].

### 2.1. Sensor Tag Concept

As in a lot of production environments, automatically placing stickers to products is a well-established process, and sensor design is based on a sticker-based sensor tag approach to facilitate automated sensor placement in an industrial context. To make this possible, the sensor tag needs to be flexible, thin and sufficiently small, which are properties that simultaneously are necessary for material integration. To withstand the conditions present during fabrication, the sensor tag needs to be sufficiently resilient to elevated temperatures. For that reason, all components were selected to have rated operating temperatures higher than or equal to 125 °C. As the tag is meant to operate without either wires or a battery, all power needs to be harvested wirelessly via the reader-supplied electromagnetic field. Consequently, a very low-power design of the electronic circuitry is necessary. Moreover, to ensure broad compatibility and easy usage, a widely used RFID standard is employed (ISO 14443, [6]). Simultaneously, this makes possible using off-the-shelf components for wireless communication, in turn simplifying the design and reducing cost. As economic producibility of the sensor tag is targeted, standardized components are also used for the electronics, including several commercially available Integrated Circuits (ICs) to reduce size, power consumption and to simplify circuit layout. The resulting concept for the sensor circuitry is displayed in Figure 2 and will be described in the following paragraphs.

### 2.2. RFID-Transponder and Micro Controller

Tag circuitry is built around an IC (NHS3152 by NXP Semiconductors, ref. [44]) combining both a micro-controller and an RFID transponder in a single package, which handles wireless communication and power supply of the sensor circuitry by harvesting power from the RFID field via a resonant circuit. The latter consists of the antenna coil $L_{ant}$ and an internal capacitor $C_{int}$ in the transponder, resonating at the frequency of the electromagnetic field output by the reader, in this case 13.56 MHz. The harvested power is output by a special General Purpose Input Output (GPIO) pin of the transponder, which, according to the transponder datasheet [44], can drive up to 20 mA depending on the available power. This voltage $U_{0}$ is stabilized via several ceramic capacitors $C_{B}$ for subsequent circuitry. The nominal output voltage of the GPIO pins is 1.8 V, though this is influenced by the tag-reader distance. To give feedback to the user, an LED is included to indicate tag power status. The integrated micro-controller (ARM Cortex M0) collects measurement data from the temperature sensor and the strain gauge, and handles wireless communication via the integrated RFID interface.

#### Sensor Elements

For strain measurement, a 4 mm × 4 mm, full-bridge strain gauge (S5020 by Micro Measurements, [45]) with four active gauges is selected, to ensure intrinsic temperature compensation and to provide for high sensitivity.

It is specified up to an operating temperature of 205 °C and provides a gauge factor of 2.1. To enable bidirectional strain measurement, two gauges each are placed perpendicular to each other, as illustrated in Figure 3a, resulting in an increasing bridge voltage dU for strain in *Y*-direction, and a decreasing bridge voltage for strain in *X*-direction. This gauge configuration principally results in a lateral contraction (νϵ, occurring perpendicular to the component strain with ν being Poisson’s ratio of the material) being superimposed on the bridge signal, which needs to be taken into account during the calculation of the actual mechanical strain. In this case, the lateral contraction increases the bridge output, as a contraction perpendicular to the main strain direction influences the bridge output the same way as the actual strain itself, thereby making the overall sensor more sensitive.

An Instrumentation Amplifier (INA) (INA333 by Texas Instruments, ref. [46]) is then used for amplification of the resulting bridge voltage. Amplification gain is set by an external resistor $R_{G}$. Bridge voltage is filtered pre-amplification by a 100 Hz (−3 dB) R-C low-pass filter to reduce noise. To offset the amplified voltage for digitization, a reference voltage obtained via a voltage divider ($R_{1/2}$) is added via the reference input of the INA, as the subsequent Analog to Digital Converter 420 (ADC) uses a single-ended configuration.

As the input range of the ADC is GND to $U_{0}$, this setup minimizes the measurement error caused by fluctuations of the supply voltage $U_{0}$ due to bridge voltage, reference voltage and the comparison voltage used by the ADC all being influenced the same way, similar to a common-mode noise reduction concept. As fluctuations in supply are likely, due to wireless power transmission and possibly varying distance to reader, this concept is necessary. Both bridge and reference voltages are therefore automatically centered between the ground and supply voltage, ensuring that fluctuation of supply least influences the measurement precision. Digitization is conducted via a 14 Bit, small footprint, low power ADC (ADS7052 by Texas Instruments [47]), which communicates with the micro-controller via Serial Peripheral Interface (SPI).

For measurement of the matrix temperature, a digital, low-power temperature sensor (TMP117 by Texas Instruments) is used [48], which is connected to the micro-controller via an Inter-Integrated Circuit (I2C) Bus. Maximum sensor error is guaranteed to be ±0.2 °C for −40 °C to 100 °C and ±0.3 °C for −55 °C to 150 °C. It was chosen for its small footprint and high accuracy, as it was designed to replace standard Pt-100 thermistors. Moreover, it is factory calibrated, ensuring accurate temperature readings over its full operating range of −55–150 °C.

In summary, the system presented aims to provide the following advantages:Completely wireless and battery-less principle: no wires complicating vacuum setup and later-on part handling, long lifetimeSensor usage throughout the whole FRP part life cycle.High precision, material-integrated temperature measurement, therefore giving higher information depth than surface based measurements, facilitating FRP production process monitoring and optimization.Biaxial, temperature-compensated strain measurement for FRP load measurement and structural health monitoring.Additional advantages of material-integrated “intelligence”: material integrated storage for part identification, maintenance or measurement data“Sensor tag” concept facilitates automated sensor placement during FRP production.

### 2.3. Sensor Fabrication

After sensor conceptualization and initial circuit tests on a breadboard, the sensor substrate, i.e., a Flexible Printed Circuit Board (FPCB) is designed, manufactured and assembled with all the necessary components.

#### 2.3.1. Sensor Substrate

For the PCB material, polyimide is chosen, as it is currently one of the industry standard materials for production of FPCBs, allowing for easy and cost-effective fabrication of the sensor system. The FPCB was fabricated by a commercial PCB-manufacturer, resulting in a 180 µm, double-layer copper–polyimide board (see Figure 4). Following the sensor tag approach, the sensor substrate is designed to be rectangular to ensure a sticker-like behavior and applicability to common sticker placement techniques, while also allowing for the easy distinguishability of the sensor orientation, which is relevant as it corresponds to the intended strain measurement direction.

To reduce mechanical tension around the substrate edges, its corners were designed to be rounded off, according to the findings in [29]. As is also shown in [29], the mechanical wound effect, i.e., negative effect of a material-integrated inlay on the surrounding component’s structural integrity, is minimized when the Young’s Modulus of the inlay is smaller or equal to that of the surrounding material. As polyimide has a Young’s modulus of around 3.1 GPa [49] and the Young’s Modulus of cured matrix materials (e.g., epoxy resin) is usually higher (e.g., 5.3 GPa for Hexion RIMR135/RIMH137, [50]) this design condition is satisfied for the material chosen.

#### 2.3.2. Antenna

The antenna coil for the transfer of energy and data is designed as a spiral copper trace, framing the PCB on the outside (see Figure 4). Sensor circuitry is placed in the center of the tag, so that the antenna encircles the electronics. A minimum distance of 4 mm was kept between the inner coil winding and the next metal structure in the center for best antenna performance, as recommended in [51]. To optimize resin flow through and around the sensor, holes were designed in the sensor substrate, especially in between the antenna and the electronics. This allows the resin to flow through during infusion, thereby creating resin bridges in the cured structure, increasing stability and possibly reducing the potential for delamination around the sensor. To minimize the wound effect created by the tag, small package sizes were chosen for the electrical components (e.g., 0402 for passive components). Package size was limited though by the necessity of manual soldering and assembly. For future versions, further size reduction is possibly by decreasing the sizes of the electronic components.

#### 2.3.3. Assembly

For assembling the sensor system, the electronic components and ICs were soldered to the PCB with manually dispensed soldering paste. A fully assembled tag is displayed in Figure 4. At its thickest point, which is where the micro-controller is mounted, it has a maximum total thickness of 1.2 mm, while the tag itself has a thickness of ca. 195 µm for both copper layers present.

The strain gauge was mounted to the tag surface with dedicated epoxy-based strain gauge glue supplied by the strain gauge manufacturer, to ensure the best mechanical coupling and minimal adhesion degradation over time. To connect the strain gauge to the electronics, the connection pads of the gauge and the PCB, which are directly adjacent, were bridged with conductive glue. This is to allow for some connection flexibility, to not influence strain measurement and to achieve mechanical insensitivity and a low profile (e.g., as compared to protruding wire bonds that would possibly break during tag handling). A close-up of the glued connection is shown in Figure 3b. The strain gauge is symmetrically positioned on the *Y*-axis of the PCB, with maximum distance to the antenna or other components, to minimize mechanical influence on the gauge itself by adjacent structures. Also, to that end, no components were placed on the PCB along the measurement axes, and holes were placed directly around the gauge to improve the mechanical coupling to the surrounding matrix material.

Apart from the original sensor tag prototype, a smaller version comprising a miniaturized antenna was built, to see if reduction in overall footprint is possible, which is displayed in Figure 5b. The miniature version did function as expected but reading range was reduced to about 21 mm.

## 3. Experiments with the Sensor

This section reports in detail the methods and experiments conducted with the sensor. Measurement setups are described for fabrication monitoring experiments and for mechanical tests with sensor-integrated specimens. For the investigation of the system limitations, two experiments are presented that are designed to find the maximum reading distance and limits of the operational temperature. Results of the experiments described are given in Section 4.

### 3.1. Fabrication Monitoring Setup

In order to investigate the sensor functionality for FRP fabrication monitoring, sensor tag integration was tested in two experiments [52], to ascertain the correct sensor function and communication, despite the conditions present inside the laminate during infusion and curing. Two separate FRP boards were fabricated during the experiments conducted, using a VARI process.

The process setup and sensor position inside the laminate is schematically displayed in Figure 6.For each experiment, three sensors tags each were placed inside the eight-layer glass fiber layup. Reference temperature sensors were placed for comparison, one in the resin reservoir and one in the center of the top vacuum foil (see Figure 7). The sensor tags were positioned beneath the second layer from the top, with specific spacing relative to each other and the textile edges (materials used: triaxial glass fiber textile (0°, ±45°, 854 g/m^2^), (matrix: Epikote RIMR 135 + RIMH137). A triaxial fiber orientation was used to create a component with anisotropic mechanical properties for testing of the sensor function, and also as this is one of the common configurations used in structural parts (e.g., wind energy rotor blades, [53]). A schematic representation of fiber orientations is given in Figure 8. The matrix material was chosen as it is a widely used combination of resin and hardener and can also be used for structural parts [50].

### 3.2. Strain Sensor Characterization Setup

Prior to the experiments, the resin and curing agent were thoroughly mixed and degassed in a vacuum chamber, to prevent formation of bubbles which would create voids in the resulting composite. To ensure no degradation of adhesion, sensor tag surfaces were cleaned and degreased with isopropanol. To fix the sensor tags in position on the textile, they were attached with an epoxy resin-based glue [54], that is specifically designed for usage in FRP laminate, as it dissolves in the matrix material during infusion and curing.

Before infusion was started, the table temperature $T_{R}$ was set to 40 °C while the resin was heated to ca. 30 °C in a water bath. Opening of the infusion valves marks the start of the experiments (i.e., t = 0 s). Relevant events during infusion could visually be recorded for later correlation to the measurement data, as all materials used were transparent. Respective events are indicated in the temperature graphs (see Figure 9).

The fluid resin reached the distributing omega profile after 1:05 min, marking the actual start of infusion. Visual arrival times of the flow front were recorded for each sensor tag; they are presented in the resulting measurement diagrams. After approximately 17:00 min, the whole mold was visibly filled. To ensure complete fill-up of the mold, the resin inlet valve was closed three minutes later. At 22:00 min, the whole setup was covered with an insulating fleece. This was done to reduce heat dissipation via the surface to create a more homogeneous distribution of temperature throughout the setup and thereby accomplish consistent cure over time.

For curing, the table temperature $T_{R}$ was subsequently increased. First, to initiate the curing reaction, $T_{R}$ was set to 55 °C. After 2:12 h, $T_{R}$ was increased to 65 °C to speed up the reaction of the remaining components. After another half hour, at 2:32 h, $T_{R}$ was again increased to 75 °C in order to ensure a full and consistent cure throughout the specimen. The main reason for this stepped temperature profile was to prevent local overheating by the exothermic reaction. During the whole production process, all sensors were wirelessly read out via an RFID reader [55] each, which were mounted above the tags (see Figure 7). Reader-tag distance was set to 20 mm, while spacing between the tags was 100 mm.

To examine the correct function and accuracy of the strain sensor, bending tests were conducted with sensor-integrated FRP samples. The specimens were cut from the FRP boards made in the prior fabrication monitoring experiment (see Section 3.1). One of the used FRP specimens is shown in Figure 10b. Specimen geometry was chosen so that symmetrical loading in both *X* (longitudinal) and *Y* (transversal) directions could be achieved (see Figure 10b for coordinate system definition). For that reason, the strain gauge was positioned in the very center of the quadratic specimen. Additionally, bending was maximized at the position of the strain gauge.

Strain sensor characterization was conducted via a testing machine, using a three-point-bending test configuration. Measurement data for both the sensor tag and reference measurement were recorded via a Programmable Logic Controller (PLC), that was connected both to the testing machine and the RFID reader. This way, time-synchronized recording is possible for both the actual (i.e., reference strain) measured by the testing machine, and the sensor data wirelessly read by the RFID reader, eliminating the need for later data correlation.

Six sensor-integrated specimens were examined in this experiment. For bending, they were positioned in a way that the bending axis was parallel to the measurement direction to be examined, with the force being directly applied in the center of the tag. To achieve strain at the layer of the tag and to not negatively influence the electronics, specimens were placed with the tag facing down, i.e., away from the cylinder applying the force, and towards the reader (see Figure 10a). The corresponding tag-reader distance was set to 20 mm. Specimens were deflected with a constant speed of 0.01 mm/min, until a maximum force of 1 kN was measured by the testing machine. At this position, the machine stopped further movement and held the position for 5 s, after which deflection was reversed with the same constant retraction speed. This procedure was repeated six times for each specimen and for both *X* and *Y* direction. Strain sensor characterization results are presented in Section 4.2.

### 3.3. Reading Distance Experiment

To investigate the maximum reading distance of the sensor system, experiments with varying tag-reader distances were carried out for both loose and material-integrated sensor tags. To see if the FRP presence influences the reading distance and transmission of energy and data, the thickness of the FRP material between the reader and sensor was varied as well.

Three tags each were placed on a glass surface beneath an RFID reader that was fixed in position by a laboratory stand. In order to least influence the electromagnetic field, the distance of the glass surface to the supporting fixture was set to ca. 25 mm by choosing a large Petri dish with a high brim and positioning it upside down on the supporting fixture. Figure 11 schematically displays the setup. The reader-tag distance (d) could be adjusted by setting the fixture height (h) via a screw attached to the supporting fixture. The reader-tag-distance was controlled manually using a caliper. For investigation of the effect of FRP presence, material thickness in between the reader and tag was increased by sequentially stacking FRP plates on top of the specimen under test. Results of the experiments are presented in Section 5.1.

## 4. Characterization and Results

This section gives the results for the experiments conducted with the sensor. First, fabrication monitoring trials are described. Subsequently, mechanical tests for characterization of the strain sensor and sensor tag influence on FRP strength are presented. In the last section of this chapter, system limitations in terms of maximum temperature and reading range are examined. For a description of the respective measurement setups, see Section 3.

### 4.1. Fabrication Monitoring Results

The experiments on fabrication monitoring and sensor tag integration give an insight in the possibilities to track resin flow front and monitor resin cure via the sensor tags presented. Moreover, the significance of the material-integrated temperature measurement in relation to surface-based measurements is investigated by comparison with a temperature sensor on top of the layup for reference.

For both experiments, all sensor tags remained functional and were communicating measurement data throughout the whole process and after. Temperature measurements are illustrated below. Figure 9 shows a detailed overview of the temperatures recorded during infusion, while Figure 12 displays the entire process until cool down. Apart from some few random reading errors for tags two and three that could be observed from 2:12 h to 4:02 h, all communication worked as expected. These reading errors could not be reproduced later on at similar and higher temperatures, which leads to the conclusion that the errors were not due to the elevated curing temperatures.

Regarding measurement differences in between tag values, a maximum deviation of approximately 1.9 °C for board one and 2.1 °C for board two could be read, respectively. This deviation was in both cases calculated for the two tags positioned at the greatest relative distance, e.g., the first and the last. Consequently, it is assumed that the deviations were mostly caused by inhomogenous heat distribution over the heated table surface. This conclusion could later be validated by thermal imaging. To remove outliers due to reading errors mentioned above, all data were Hampel-filtered with a window size of 5.

Advancement of the flow front through the setup was both visible from the outside and apparent in the measured data. As illustrated in Figure 9, a close correlation of sensor data and visually recorded resin arrival times is observable. When the resin reaches the sensors, the measured temperature drops distinctly, as resin temperature is lower than that of the dry, heated textile. This drop is followed by a distinct temperature increase shortly after resin arrival, which can be attributed to an increase in transported heat energy from the heated table surface to the sensors, due to higher thermal conductivity of the resin as compared to the dry textile.

Resin cure is then initiated in the first heating phase. Measurement values clearly reflect the exothermic nature of the curing reaction. Temperature measurements inside the laminate show a distinct local maximum around 1:10 h that can be attributed to the heat generated by the reaction. Despite the cover fleece, this remarkable feature is not reflected in the surface-based measurements—surface temperature stays nearly constant, showing a maximum drift of ca. 2 °C throughout the first phase of heating. Overall, surface temperature data follow sensor tag data, though the values of the reference sensor are generally smaller, e.g., temperature inside the laminate being several degrees higher. This is to be expected, as heat is dissipated into the surrounding air, in spite of the insulation fleece. This difference reduces during the two subsequent heating phases, indicating a lesser amount of heat being created inside. This could be attributed to the reduced activity of the curing reaction and therefore further progression of the cure.

### 4.2. Strain Sensor Characterization Results

This section summarizes results of the bending tests conducted, after briefly mentioning some general findings. Preliminary results on strain sensor characterization have already been partially reported in [56].

In its idle state, the strain readings of the integrated tags differed even before the first loading. This is probably caused by strain gauge deflection due to integration, e.g., bending of the gauges due to compression between the fibers of the textile. Also, even though low tolerance resistors were used for INA reference voltage, variations can possibly occur between the tags due to resistor tolerance, influencing zero strain voltage. For an actual application, this influence could be reduced by in-place calibration after integration.

For all specimens, the first loading resulted in a different force and strain response compared to subsequent loadings. Force values increased slower during bending, i.e., at larger deflection values. In other words, the specimens were deflected further before reaching the force set point of 1 kN, as displayed in Figure 13. After this initial ‘settling’ of the material, force and deflection responses were repeatable and showed no signs of fatigue behavior. For this reason, all characterization measurements are based on loading iterations 2 to 6. This effect is observable for specimens both with and without a sensor tag, as could be confirmed by later comparison, indicating a material-related effect (see also Section 4.3). As deflection was always measured from the point where the testing machine measured a force of more than 0.1 N, the deflection measurement for the first loading could have been influenced by minor deformations of the specimen surface, which did not occur for later measurements.

#### 4.2.1. Deflection and Strain Measurements

As shown in Figure 14, all FRP specimens showed generally similar behavior when loaded. Mean deflections differ for bending in transversal and longitudinal direction, which is to be expected due to anisotropic material properties. Specimens are stiffer for bending in *X*, i.e., perpendicular to the main fiber direction (see Figure 10b), resulting in approximately 0.2 mm/mm less deflection compared to bending parallel to the main fiber direction. No significant difference is found when comparing the deflections of the two different FRP boards (e.g., board I and board II), though mean deflections differ between specimens overall. Maximum deflection difference was found to be ca. 0.1 mm/m for both directions.

Figure 15 is an example of the bending measurement data for a single specimen, for both parallel and perpendicular bending. The dark lines show the mean value, while the lighter, shaded areas illustrate the standard deviation for both curves (standard deviation is very small for the reference data). It can be found that tag measurements closely follow the reference data, though with a higher noise and higher standard deviation. Measurements for bending parallel to the main fiber direction (Figure 15b) result in a longer duration, as the material is less stiff and can therefore be deflected further before reaching the force set point of 1 kN. This is also cause for bending parallel to the main fiber direction resulting in larger strain values for the same force. While maximum strain in *X* (perpendicular to main fiber direction) is ca. 1.7 mm/m, bending in *Y* (parallel to main fiber direction) results in approximately 2.3 mm/m for the same force.

In general, for both loading directions, a slightly curved, i.e., non-linear trend in sensor strain data can be observed for both increasing and decreasing deflection (see Figure 16a and Figure 17a). This is reflected to some degree in the sensor error, i.e., difference to reference, as displayed in Figure 16b and Figure 17b, which show local maxima during loading and unloading processes. After full unloading of the specimen, sensor tag strain values show a slight residual offset (see Figure 15b). This is likely caused by temporary plastic deformation of the specimen due to prior bending. For the other bending direction, this effect is probably opposed by the fiber presence. For loading perpendicular to the main fiber direction (Figure 16a), a maximum strain of 2.0 mm/m could be measured for a force of 1 kN, for specimen II_1_. Loading parallel to the main fiber direction (Figure 17a) causes a corresponding maximum strain of 2.6 mm/m, this time for specimen I_2_. Comparing different specimens, mean strain values measured by the sensor tags differ up to approximately 0.6 mm/m for both directions.

#### 4.2.2. Sensor Error

As already mentioned in the previous paragraph, it could be found that sensor error, i.e., difference between actual strain (reference), as measured by the testing machine, showed variance between specimens and also between different loading states. Sensor errors for all experiments are given in Figure 16b and Figure 17b, while mean and maximum sensor errors are given in Table 1. In general, variation in sensor error is highest for maximum loading, while it is lowest for the unloaded states.

Averaging all values for the sensor error of bending perpendicular to the main fiber direction, it can be found that during the loading and unloading phases, the sensor tags generally measured about 0.1 mm/m more strain than given by the reference. During the holding phase, mean sensor error decreases. These effects are depicted by the black, dashed curve in Figure 16b and Figure 17b.

Averaging the sensor error of all specimens over time, it can be found that for bending perpendicular to the main fiber direction, sensor tags measured about 0.09 mm/m less strain than what actually occurred. In other words, the mean strain difference of reference and sensor tag strain is about 0.09 mm (see Table 1). For bending parallel to the main fiber direction, a mean sensor error of approximately 0.05 mm/m is found. Sensor error varies with loading, though these values have only a limited significance.

The overall difference in measured strain seems to at least partly depend on the individual tag or specimen, e.g., is caused by differences in the coupling strength of the matrix and strain gauge, as sensor error is highest for the sensor tag (I_2_) measuring the overall smallest strain in *X* and the largest strain in *Y*. Possible reasons also include tolerances of the electronic components, resulting in the differing amplification of the bridge signal. This correlation can, to some degree, be inversely observed for the tags measuring the overall largest strain in *X* and smallest strain in *Y* (II_2_ and I_3_). Comparing actual deflections of the specimens, as measured by the testing machine, it can be found that not all specimens experienced the same deflection for the same amount of applied force, which contrarily indicates differences in specimen stiffness, consequently resulting in differences of occuring and measured strain.

For bending perpendicular to the main fiber direction, standard deviation is generally below 0.1 mm/m, with specimen I_2_ showing a maximum standard deviation of 0.16 mm/m. Bending parallel to fiber direction yielded similar results (see Figure 18b), though a slight increase in standard deviation is observable for all specimens during maximum loading (around t = 90 s).

### 4.3. Tag Influence on FRP Strength

To investigate the effect of tag presence on FRP strength, comparative tests were conducted with FRP samples with and without a sensor tag. To ensure comparability, only specimens from the same fabrication run, i.e., the FRP board were compared.

The same basic setup was used as for the prior bending tests, initially comparing specimen behavior for a force of 1 kN, so as to not damage the specimens. These measurements were done for bending in both directions (*X* and *Y*). Afterwards, the bending force was increased for destructive testing, to see if the tag presence influenced failure behavior. For this, instead of a force value, a target deflection was used as the set point, resulting in destruction of the specimens. Due to the destructive nature of the experiment, bending was done only perpendicular to the main fiber direction. These experiments were done with three samples each, with and without a sensor tag for comparison.

#### Results

Prior to destructive testing, non-destructive bending tests were conducted to quantify the impact of tag integration on material stiffness. Figure 19a,b illustrate mean deflection values obtained via the testing machine, i.e., the reference measurement equipment, for all specimens, in both bending directions. Each curve represents the average deflection values of bending iterations 2 to 5, as also used in Section 4.2.

As already observed during strain sensor characterization, deflection magnitude for the same force of 1 kN differs between specimens, both for specimens with and without tag. Nonetheless, a clear trend is observable when comparing deflection values for specimens with and without a tag. All specimens with an integrated sensor tag show less deflection compared to the corresponding specimens without a tag, in both specimen orientations. This shows that tag integration actually strengthens the material locally, i.e., stiffening the laminate. This might be explained by the sensor tag acting as an additional layer in the material, instead of actually disturbing the laminate. Still, the significance of this finding is limited, as experiments do not account for other influences like fatigue that would come into play for cyclic loading during application.

Comparing the deflection of different specimen orientations, results confirm the findings of the experiments conducted for strain sensor characterization. Bending perpendicular to the main fiber direction generally leads to less deflection as compared to bending parallel to the main fiber direction. This can once again be explained by the anisotropic material properties caused by the orientations of the textile layers (0°, ±45°), as fibers strengthen the material mainly in *X* direction (see Figure 10b).

Following the non-destructive tests described in the prior paragraphs, three specimens each were loaded until breakage, to see if tag integration affects FRP breaking strength. Maximum forces for all the specimens tested are given in Table 2. Overall, the destruction force data show quite a similar trend for both sensor-integrated and sensor-less specimens (see Figure 20). Mean destruction force was calculated as 10.4 kN for samples without a sensor tag, while sensor-integrated samples broke at an average of 10.0 kN. This difference is mainly due to specimen I_2_ showing a reduced breaking force compared to the other specimens. This is probably due to the fact that prior to the destructive testing experiments, it was used for experiments regarding maximum operation temperatures (see Section 5.2), possibly degrading the material properties due to extreme temperatures and inherent material aging. After the first sign of breakage, visible as a drop in the force data, it can be seen that the material is still able to bear a load, as the measured forces do not sharply drop after the initial breaking event. Regarding actual application scenarios, this effect is desirable, as the material does not fail immediately after the first overload.

After destruction, apart from fiber breakage, one major failure mechanism observed is delamination of the textile layers throughout the specimens. This occurred at layers of tag presence, but was also observable for specimens without an integrated sensor tag (see Figure 21). A correlation with sensor tag presence can therefore not be found for the experiments conducted. Still, as the number of destructive tests is small, the significance of these results is limited and further investigations seem necessary for a conclusive assessment of the tags’ mechanical influence on FRP strength.

## 5. Investigation of System Limitations

To examine the limits of the system, several experiments were conducted, which are evaluated in the following section. Regarding the limitations of the strain sensor, the absolute limit of the measurable strain was found to be approximately 9 mm/m, as discovered during the destructive bending tests described in Section 4.3. This is a result of INA output range, as output voltage of the INA is limited by design (U_0_ − 0.1 V, [46]). In practice, strain measurement for cyclic loading applications is limited by the strain gauge itself, as too high a strain can irreversibly damage the metal structure of the gauge. For this reason, it is advisable to not strain the gauge more than 1.8 mm/m for cyclic loading, according to the gauge data sheet [45].

### 5.1. Maximum Reading Distance

During the experiments on the maximum reading distance (see Section 3.3 for setup description), two distinct events were noticeable for increasing the tag-reader distance. Before the tags actually dropped out, as indicated by no data being read by the reader, it was noticeable that the measurement values did not change anymore, even though the reader was still able to read the tag memory. As the voltage induced in the antenna coil decreases with increasing distance, this effect may be explained by decreasing supply voltage on the tag. At a certain distance, supply voltage output by the RFID transponder is insufficient to power the tags measurement electronics, while the voltage induced in the antenna is still high enough for the RFID transponder to power its memory and communication functions. In this state, the last sensor value stays in the memory, as no new data are obtained via the measurement electronics. To distinguish this effect, results are divided into two categories. Figure 22 illustrates distances at which the sensor values did not change anymore, while Figure 23 shows results for the distances at which no data could be read from the transponder memory anymore. Additional to the actual distance values, a linear curve has been fitted to the data, using a linear polynomial fit, to allow for visual identification of the general trend.

Results show that the distance at which the measurement values stay constant is not significantly influenced by the FRP thickness between sensor tag and reader. Regarding the maximum reading distance, this is also true. It can therefore be concluded that the reading distance is not significantly influenced by FRP thickness.

Comparing results for embedded and separate sensor tags, both the maximum reading distance and constant values distance are approximately 2 mm higher for separate sensor tags. Integration therefore seems to slightly decrease the reading distance. An explanation for this finding is a possible deformation of the antenna due to the compression in between the fibers, leading to degradation of the inductive coupling.

### 5.2. Operating Temperature

To investigate the maximum and minimum limits for the operation temperature of the sensor system, experiments were carried out with both material-integrated and unintegrated, i.e., loose tags. Temperature was controlled via a climate chamber, in which the samples were put. To have an accurate value for temperature close to the specimens, a reference sensor was used. A Pt100 in a four-wire-configuration was employed for this, which was mounted to the specimens with heat resistance tape. Temperature varied from −40 to 130 °C, while the sensor tags were continuously read out wirelessly.

Results for both tags are displayed in Figure 24. It can be seen that for both tags, a minimum working temperature could not be determined, as temperatures below −40 °C could not be established with the available equipment. Regarding the maximum temperature, the unintegrated tag dropped out at 126.9 °C, while the tag integrated in GFRP dropped out at a temperature of 137.2 °C as measured by the reference sensor. Apart from that, it can be seen that the temperature measured by the tag closely follows the data measured by the reference. The difference visible between the sensor tag and reference data for the integrated tag might be explained by the fact that the reference, due to the integrated nature of the sensor tag, could only be applied to the surface of the FRP specimen, therefore measuring the surface temperature, as opposed to the tag inside. Even though it was placed as close to the sensor tag temperature sensor as possible, minor variations in temperature are likely. Regarding the results of the experiments, it can be found that the tag is very well able to function in the extended industrial temperature range of −40 to 125 °C.

## 6. Discussion

The sensor system presented was found to fulfill the intended purposes and requirements in the experiments conducted, thereby showing the approach to be generally feasible in a proof-of-concept context.

During the integration experiments, the sensor tags were able to track progression of the flow front and also indicate progression of the curing reaction via temperature measurement. This way, they provide a huge potential for process optimization, and saving of time, energy and resources by helping to ensure correct resin flow and monitor the cure. Due to the sensing principle, flow front monitoring is yet only applicable if a measurable difference exists between the temperatures of the fluid resin and the temperature of the textile and mold. Depending on the materials and processes used, this might be a limitation.

Regarding strain measurement capability, the system presented was able to measure the strain of the host FRP specimens in both lateral and transversal directions with usable results. Nevertheless, the potential for strain sensor optimization is apparent, especially regarding linearity. The difference in the sensor and reference values varies with the applied force and in between specimens, even though the average sensor error is below 0.06 mm/m for both directions. This might be explained several ways: possibly, differences exist between specimens in the adhesion of the tag and surrounding matrix. Moreover, as lateral contraction is superimposed on the sensor signal, actual material properties and especially non-uniformities can negatively influence sensor accuracy. Further investigations seem necessary in this regard.

Regarding tag influence on the host component, a number of experiments were conducted, and no negative impact could be found. These results have only limited significance though, as they were small in number and only bending strength was investigated. Still, optimization of tag geometry seems desirable to further reduce the tag footprint and therefore the mechanical influence on the host material. In this regard, especially a reduction in thickness would potentially be possible by using different packages for ICs, e.g., wafer level or chip scale-level packages, and smaller passive components.

The tag reading range was found to be sufficient for all experiments and therefore seems to be adequate for actual applications in the context intended. Nonetheless, the reading range could possibly be further optimized by using an even better tuned antenna and by decreasing the power consumption of the electronics. As wireless energy and data transmission are key features of the system presented, it allows for a huge simplification of sensor integration into FRP. However, this advantage also implies a restriction regarding possible applications. As energy is principally transferred by inducing a voltage in the antenna coil, the tag can only be supplied with sufficient distance to the conductive surroundings. Metal in direct vicinity reduces the reading range until no sensor readout is possible with the tag directly sitting on a metal surface. This needs to be considered during part handling and mold selection. Similarly, tag application is limited to usage with non-conductive fibers, e.g., glass, aramid or other plastics, as these do not inhibit energy transfer.

## 7. Conclusions

This paper presents a completely wireless and battery-less sensor for integration into fiber-reinforced plastics, based on a ‘sticker-like’ sensor-tag approach. After describing sensor motivation, sensor concept and design and tag fabrication, experiments were presented, showing that the system can be used for several functions during the FRP part life cycle. During FRP production, the sensor tag can be used to track local resin temperature inside the laminate, thereby allowing for monitoring of the resin flow front and progression of the resin cure, potentially enabling optimization of production processes. During FRP component application, the integrated sensor tag can be used for both structural health monitoring applications and ‘smart component’ functionality, via the integrated strain sensor and micro-controller. As the system uses standardized electronic components and standardized wireless communication, it is both cost-effective and widely compatible to RFID reading hardware. Bidirectional strain measurement is possible via the sensor tag, and could be found to have an average error of less than 0.06 mm/m, though sensor error behavior is nonlinear without further calibration. Regarding system limitations, an averaged maximum reading distance of at least 38 mm could be measured, while the operational temperature range was found to be at least −40 to 126.9 °C. Regarding the mechanical impact on FRP structural integrity, no negative impact could be found, but further research is necessary to investigate integration effects in detail.

## Figures and Tables

**Figure 1 sensors-23-06375-f001:**
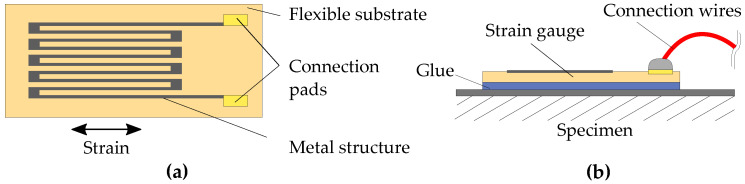
Schematic representation of (**a**) strain gauge principle; (**b**) strain gauge application to a specimen surface.

**Figure 2 sensors-23-06375-f002:**
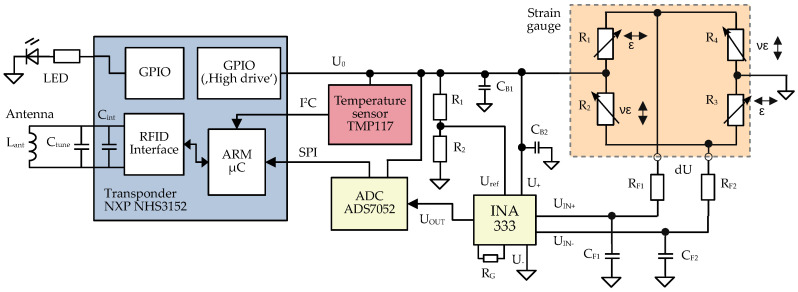
Schematic representation of the sensor tag circuit, adapted from [43]—prior version first published in *Advances in System-Integrated Intelligence*, pp. 182–193, 2022 by Springer Nature.

**Figure 3 sensors-23-06375-f003:**
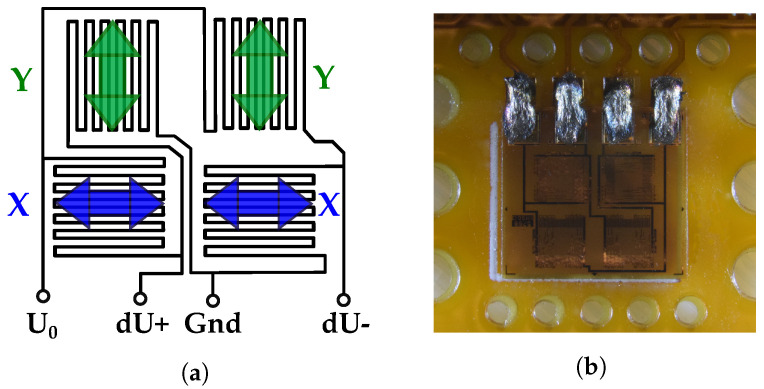
(**a**) Schematic representation of the full bridge strain gauge configuration used for a biaxial strain measurement [45]. (**b**) Strain gauge glued to sensor substrate, electrically connected with conductive glue.

**Figure 4 sensors-23-06375-f004:**
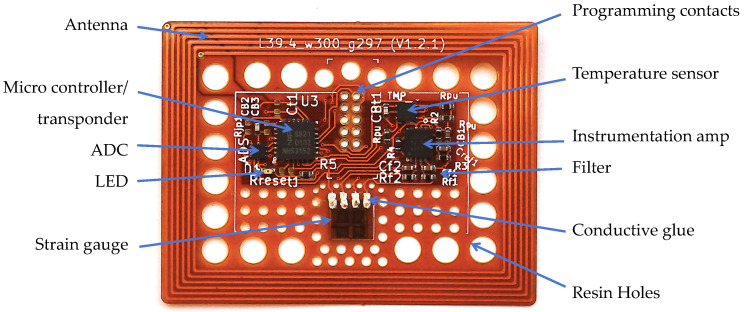
Fully assembled sensor tag with component descriptions.

**Figure 5 sensors-23-06375-f005:**
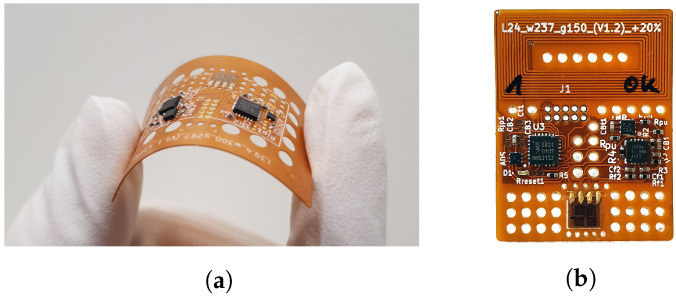
(**a**) Fully fabricated, flexible sensor tag. (**b**) Smaller version prototype of sensor tag (not to scale with left image).

**Figure 6 sensors-23-06375-f006:**
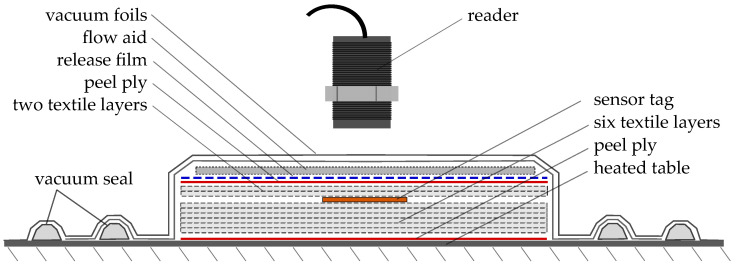
Laminate layup and experiment concept in fabrication monitoring experiment.

**Figure 7 sensors-23-06375-f007:**
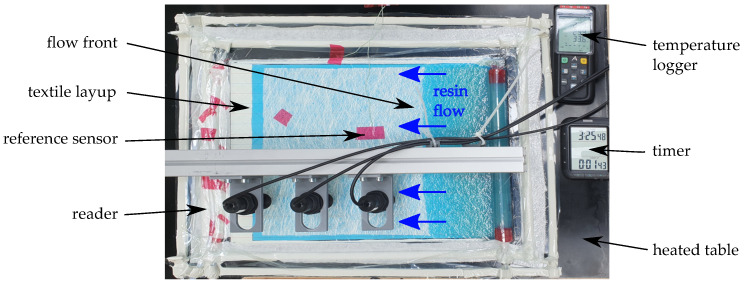
FRP fabrication experiment for sensor tag integration.

**Figure 8 sensors-23-06375-f008:**
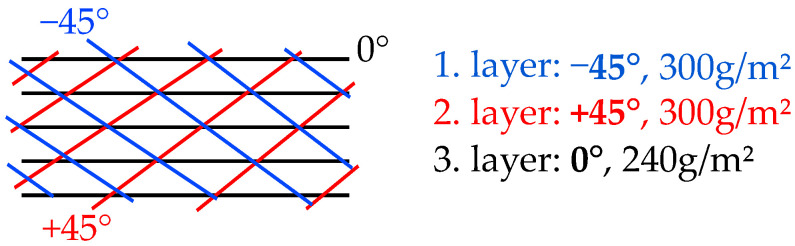
Fiber orientations used for fabrication monitoring experiments.

**Figure 9 sensors-23-06375-f009:**
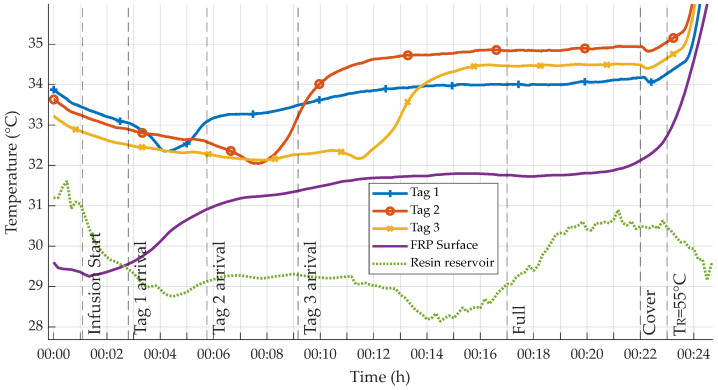
Sensor tag and reference temperature measurements during infusion. Visible arrival times of resin flow front marked by dashed lines.

**Figure 10 sensors-23-06375-f010:**
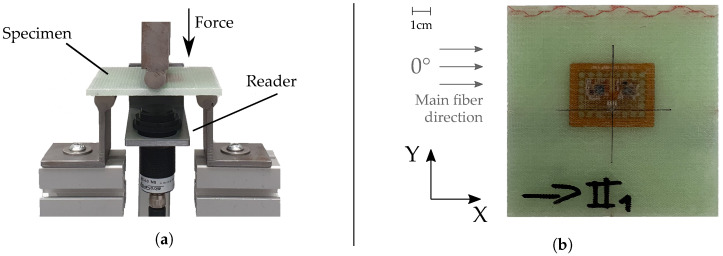
(**a**) 3-Point-bending test setup for mechanical experiments. (**b**) Sensor integrated FRP specimen for bending tests, 0° fiber direction indicated by arrow.

**Figure 11 sensors-23-06375-f011:**
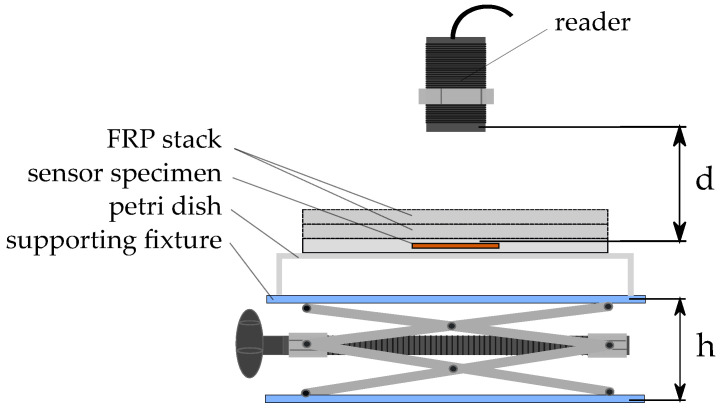
Schematic illustration of distance measurement setup. Setting of d possible by changing h via screw.

**Figure 12 sensors-23-06375-f012:**
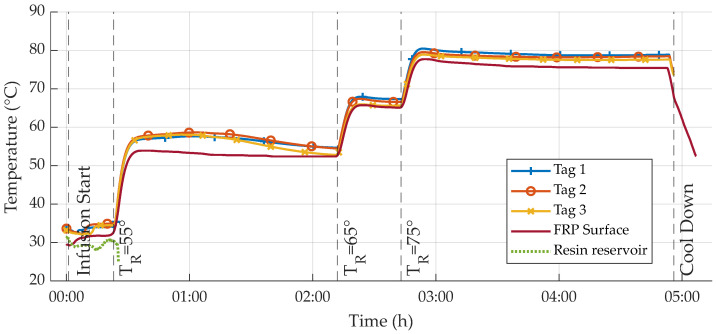
Sensor tag and reference temperature measurements for the whole production process. Changes in heating set points indicated over time by dashed lines.

**Figure 13 sensors-23-06375-f013:**
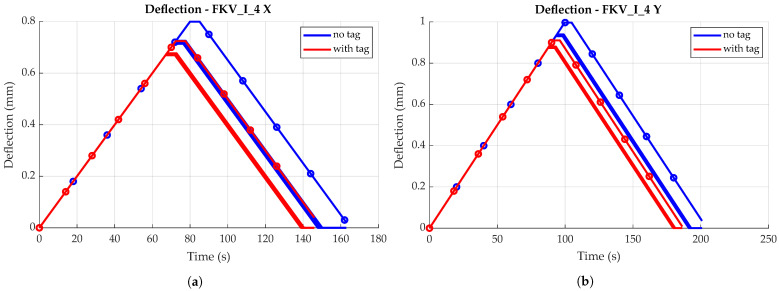
Deflection graphs for FRP specimen I_4_ as measured by testing machine, comparing deflections with and without tag. First deflection yields different behavior for each direction (circle markers). (**a**) Deflections for bending perpendicular to main fiber direction. (**b**) Deflections for bending parallel to main fiber direction.

**Figure 14 sensors-23-06375-f014:**
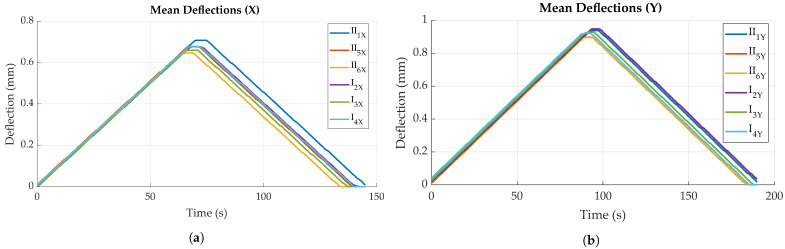
(**a**) Averaged deflections for bending perpendicular to main fiber direction, *n* = 5. (**b**) Averaged deflections for bending parallel to main fiber direction, *n* = 5.

**Figure 15 sensors-23-06375-f015:**
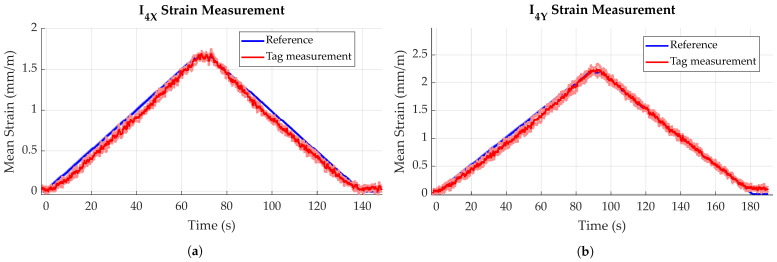
Mean strain data for FRP specimen I_4_ compared to reference, *n* = 5. Standard deviations indicated by shaded areas. (**a**) Deflections for bending *perpendicular* to main fiber direction. (**b**) Deflections for bending *parallel* to main fiber direction.

**Figure 16 sensors-23-06375-f016:**
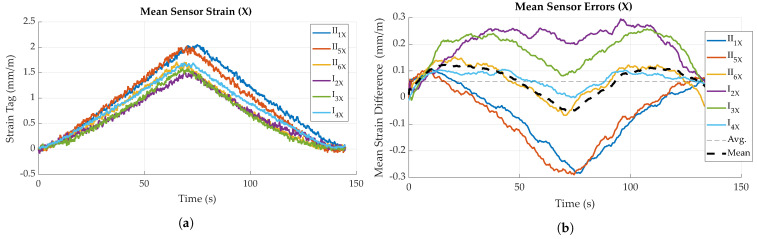
Bending perpendicular to the main fiber direction: (**a**) Averaged strain sensor readings, *n* = 5. (**b**) Averaged sensor errors (difference to reference), *n* = 5.

**Figure 17 sensors-23-06375-f017:**
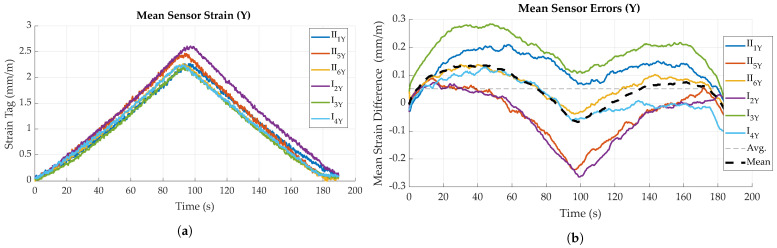
Bending parallel to main fiber direction: (**a**) Averaged strain sensor readings, *n* = 5. (**b**) Averaged sensor errors (difference to reference), *n* = 5.

**Figure 18 sensors-23-06375-f018:**
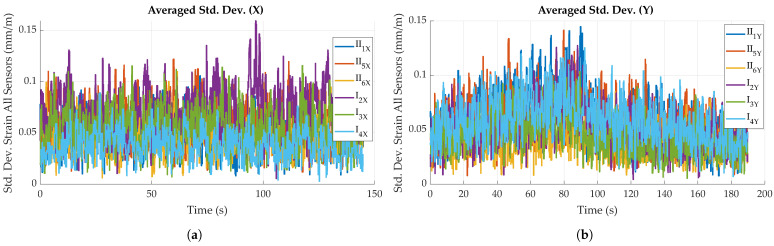
(**a**) Mean standard deviations for bending perpendicular to main fiber direction, *n* = 5. (**b**) Mean standard deviations for bending parallel to main fiber direction, *n* = 5.

**Figure 19 sensors-23-06375-f019:**
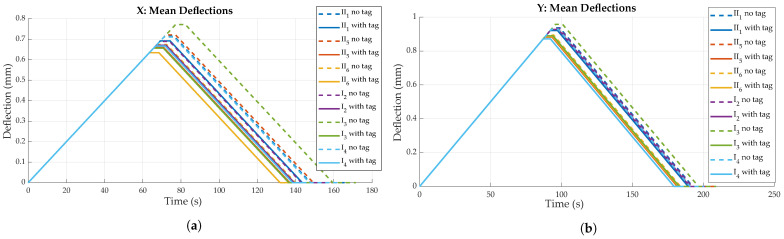
(**a**) Mean deflections for all specimens, comparing specimens with a tag (solid lines) and without a tag (dashed lines). Bending count for each curve: *n* = 5. (**a**) Data for bending perpendicular to main fiber direction. (**b**) Data for bending parallel to main fiber direction.

**Figure 20 sensors-23-06375-f020:**
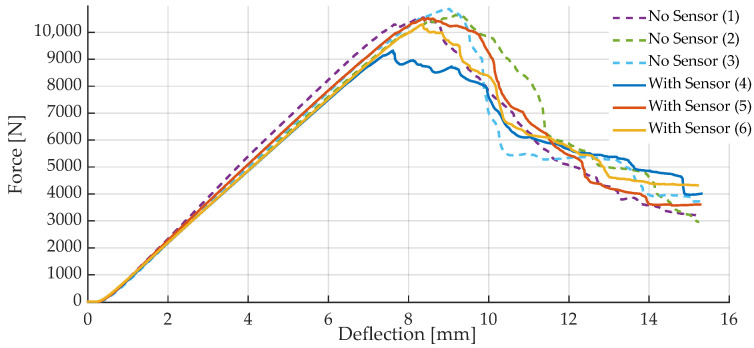
Exerted force during destructive tests for specimens with (dashed lines) and without (solid lines) sensor tags.

**Figure 21 sensors-23-06375-f021:**
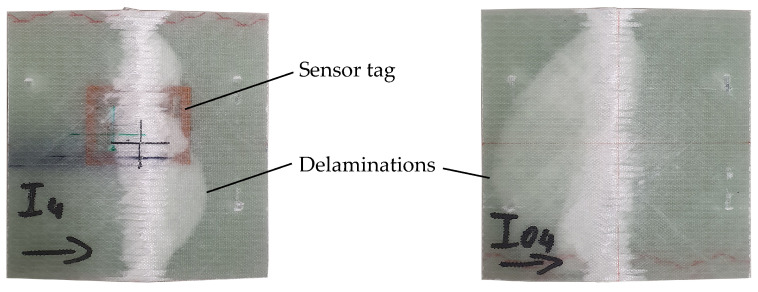
Failure comparison for FRP specimen with and without sensor tag. Both specimen show delamination close to the top layer.

**Figure 22 sensors-23-06375-f022:**
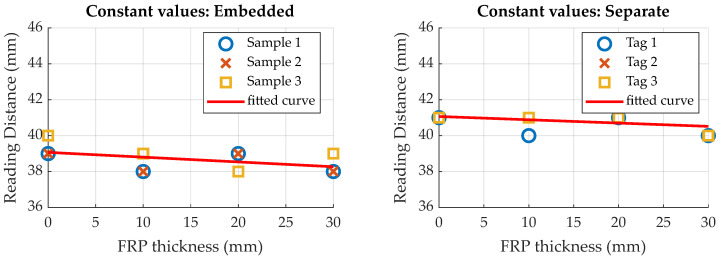
Distance at which sensor values did not change anymore vs. FRP thickness for integrated and unintegrated sensor tags.

**Figure 23 sensors-23-06375-f023:**
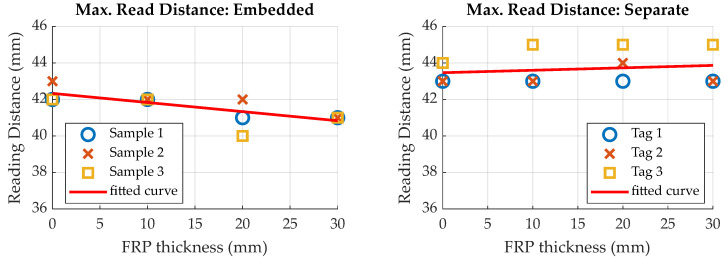
Maximum reading distance vs. FRP thickness for integrated and unintegrated sensor tags.

**Figure 24 sensors-23-06375-f024:**
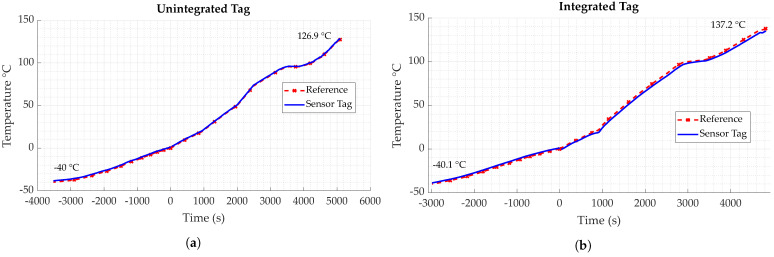
Minimum and maximum operational temperatures. (**a**) single, unintegrated sensor tag (**b**) sensor tag integrated into FRP specimen.

**Table 1 sensors-23-06375-t001:** Maximum sensor errors for the bending tests, as compared to reference.

Error	*X* (mm/m)	*Y* (mm/m)
Average error (overall)	+0.059	+0.049
Max. positive	+0.290	+0.284
Max. negative	−0.290	−0.264

**Table 2 sensors-23-06375-t002:** Maximum forces before breakage of FRP specimens.

Specimen	Max. Force (kN)	Specimen Nr.
No Sensor (1)	10.3 kN	I_02_
No Sensor (2)	10.4 kN	I_03_
No Sensor (3)	10.5 kN	I_04_
With Sensor (4)	9.2 kN	I_2_
With Sensor (5)	10.4 kN	I_3_
With Sensor (6)	10.3 kN	I_4_

## Data Availability

The raw data of the experiments can be requested from the authors.

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
