# Peer review of "Wireless, Material-Integrated Sensors for Strain and Temperature Measurement in Glass Fibre Reinforced Composites"

_sensors, 2023, doi:10.3390/s23146375_

Round 1

Reviewer 1 Report

The research on strain and temperature monitoring of FRP are rich, interesting, and systematic. The authors have innovatively proposed a new sensor and corresponding preparation method, and the monitoring results and discussion of FRP are also rich and detailed. Overall, the paper has been well designed with some innovative results and findings. Furthermore, the following comments can be considered for further improvement.

1. The writing of the abstract should include some important results and findings of this paper, such as the effect of the sensors on performance of GFRP, as well as related monitoring accuracy and results. Please also provide some potential application prospects of the research achievement.

2. In addition to higher specific strength and stiffness, the another advantages of FRP compared to metal materials are their excellent durability, fatigue resistance, and creep resistance, especially in complex environments. When considering whole life cycle, FRP and its related engineering structures have higher longevity and better durability, as well as economic costs. It is recommended that the authors consider the above comments and supplement and enrich them by reviewing the relevant research below: Polymers, 2023, 15: 2483. Materials Today: Proceedings, 2021, 46: 555-561.

3. The authors have provided a detailed summary of different monitoring methods during the production and loading process. Furthermore, please provide a simple summary of advantages and disadvantages of different monitoring methods.

4. Please break down Part 3 characterization and results into two parts. It is suggested to put the characterization methods into Part 2 (sensor design, fabrication and characterization). Part 3 should focus on results and discussion (fabrication and loading monitoring results).

5. What are the advantages of the sensor designed in this paper compared to other researchers’ sensors or general commercial sensors? Please further provide some innovative points in the design of the sensor.

6. How to evaluate the accuracy of temperature and strain monitoring presented in Figures 8-16? Suggest adding relevant analysis.

7. Is the monitoring method proposed in this paper still effective when FRP exposed to complex environmental, temperatures and loads? If the corroded medium enter into FRP, will it have the effect on the monitoring results?

8. From Figure 18 and Table 2, it can be seen that the presence or absence of sensors has little effect on the ultimate bearing capacity of FRP at fracture, and the maximum bearing capacity is only about 10 kN. When considering the high bearing capacity of FRP, will this effect be further amplified owing to the imperfect interface bonding performance between the sensor and the fiber/resin interface.

9. Please provide some important quantitative results related to this paper in the conclusion section.

Author Response

Dear reviewer,

thank you very much for reviewing the article and giving highly detailed feedback. I will gladly answer the aspects mentioned inline with the comments:

1. The writing of the abstract should include some important results and findings of this paper, such as the effect of the sensors on performance of GFRP, as well as related monitoring accuracy and results. Please also provide some potential application prospects of the research achievement.

  • I added information on applications, sensor error, system limitations and findings of integrtaion influence to the abstract.

2. In addition to higher specific strength and stiffness, the another advantages of FRP compared to metal materials are their excellent durability, fatigue resistance, and creep resistance, especially in complex environments. When considering whole life cycle, FRP and its related engineering structures have higher longevity and better durability, as well as economic costs. It is recommended that the authors consider the above comments and supplement and enrich them by reviewing the relevant research below: Polymers, 2023, 15: 2483. Materials Today: Proceedings, 2021, 46: 555-561.

  • Thank you for the recommendation. Unfortunately I am unable to access the second document. But I did include the first one and found some other publiucations on that topic, which I also included.

3. The authors have provided a detailed summary of different monitoring methods during the production and loading process. Furthermore, please provide a simple summary of advantages and disadvantages of different monitoring methods.

  • Advantages and disadvantages of the different technologies are given in Section 1.3.3 and 1.4.1-1.4.2. I am hesitant to add even more text on repeating this, as Section 1 is already very long. This is something the other referee fundaamentally critizised, so I would suggest to not include any more text on this for the sake of overall length.

4. Please break down Part 3 characterization and results into two parts. It is suggested to put the characterization methods into Part 2 (sensor design, fabrication and characterization). Part 3 should focus on results and discussion (fabrication and loading monitoring results).

  • Okay, I seperated experiment setup and results – section 3 now describes the setups, while section 4 focuses on the results, while referencing section 3 for details on the setups.

5. What are the advantages of the sensor designed in this paper compared to other researchers’ sensors or general commercial sensors? Please further provide some innovative points in the design of the sensor.

  • I added a bullet list summarizing the innovative aspects and advantageous properties in section 2.2.

6. How to evaluate the accuracy of temperature and strain monitoring presented in Figures 8-16? Suggest adding relevant analysis.

  • Temperature sensor accuracy is specified and guaranteed in / by IC data sheet. I added a sentence in section 2.2.1, giving details on sensor accuracy

  • Accuracy of the strain sensor is displayed in Fig. 16b and 16b by giving the difference between sensor reading and reference strain, as exerted by the testing machine, while mean and maximum sensor error are given in Table 1. Is there anything that you would suggest adding for further analysis of sensor accuracy? Statistically speaking, I also included standard deviation of strain sensor data in Figure 18a and 18b.

7. Is the monitoring method proposed in this paper still effective when FRP exposed to complex environmental, temperatures and loads? If the corroded medium enter into FRP, will it have the effect on the monitoring results?

  • Long term experiments and influence of extreme conditions (apart from very low to very high temperatures) have not yet been tested. However, as the sensor is integrated into the material itself, it can be assumed, that it will be protected from environmental influences as long as the FRP itself stays intact. If the FRP degrades and lets in water for example, this will probably effect the sensor in the long run. But in this case, the part itself should be checked and replaced or repaired for safety reasons anyway.

8. From Figure 18 and Table 2, it can be seen that the presence or absence of sensors has little effect on the ultimate bearing capacity of FRP at fracture, and the maximum bearing capacity is only about 10 kN. When considering the high bearing capacity of FRP, will this effect be further amplified owing to the imperfect interface bonding performance between the sensor and the fiber/resin interface.

  • Regarding the limited amount of tests we conducted, we could not find a negative influence of the tag-matrix interface on breaking strength, but this is of course no final answer / conclusive statement. As with the other experiments I therefore included a sentence in section 4.3.1 on the limited significance of the findings on mechanical influence of tag integration.

9. Please provide some important quantitative results related to this paper in the conclusion section.

  • Okay, I included details on measurement error and system limitations in the conclusion.

Once again thank you for taking the time to review the article. I am looking forward to your feedback on the new version!

Reviewer 2 Report

The article is well written and the findings are adequately explained. However, there are some comments that need to be considered in order for the article to be accepted for publication:
1. An incredible amount of general information that needs to be significantly reduced. Not only is it not helpful, it actually hinders the understanding of what the authors actually did in this study.
2. The polymers chosen for the sensor substrate must be described in detail. The reason for their choice is explained. How was the composite made? It would be good to show the structure of the composite so that the character of the fibre orientations can be confirmed.
3. Figure 23. It needs to be clarified what the difference is between the results for case A and B?

Author Response

Dear reviewer,

thank you very much for taking the time for reviewing the article and providing feedback on it. I will gladly answer the aspects mentioned inline with the comments:

1. An incredible amount of general information that needs to be significantly reduced. Not only is it not helpful, it actually hinders the understanding of what the authors actually did in this study.

  • I totally understand your remarks about length of the introduction, it is indeed quite long. I will briefly explain why I deemed it necessary. One of the reasons for this is the fact, that the reasons for designing the presented sensor system are also quite manifold, i.e. it targets to solve a multitude of problems at once / in a single system. This in turn makes necessary to introduce the problems i.e. the several motivational aspects in the introduction.
    Nonetheless, I compressed section one as far as I found possible without loosing major aspects.
    To improve readability, I also inserted a „Structure of this work“-paragraph right after the intital introduction, instead of giving the „conducted work“ section at the end of the introduction / section one. It gives a brief overview of the structure and the content of the paper in general, so that a reader can also choose to skip the background information.
    Also, the other referee explicitly highlighted detailed background information as a positive thing. I would therefore propose this as a compromise between the two opinions regarding length of section one.

2. The polymers chosen for the sensor substrate must be described in detail. The reason for their choice is explained. How was the composite made? It would be good to show the structure of the composite so that the character of the fibre orientations can be confirmed.

  • I included an explanatory figure on fiber orientation in the description of the fabrication monitoring setup (Section 3.1, Figure 6)

3. Figure 23. It needs to be clarified what the difference is between the results for case A and B?

  • Thank you for the remark, the caption was indeed not well understandable. Figure 23a shows results for the lose, i.e. single sensor tag in air, while Figure 23b shows results for a sensor tag integrated into an FRP specimen. I changed the caption to be easier to read.

In terms of document structure, you will find that the overall sequence of experiments description and results changed somewhat. The other referee told me to split the respective setup and results parts into two different sections. I hope that this also improves readability of the whole document.

Additionally, I improved the respective section introductions and some subheading, so that it is clearer what each section actually presents, aiding the reader in following document structure and argumentation.

Once again thank you for taking the time to review the article. I am looking forward to your feedback on the new version!

Round 2

Reviewer 1 Report

Accepted.